# The drivers of West Nile virus human illness in the Chicago, Illinois, USA area: Fine scale dynamic effects of weather, mosquito infection, social, and biological conditions

**Surendra Karki, William M. Brown, John Uelmen, Marilyn O'Hara Ruiz, Rebecca Lee Smith**[ID]*

Department of Pathobiology, University of Illinois, Urbana-Champaign, Urbana, Illinois, United States of America

* rlsdvm@illinois.edu

## Abstract

West Nile virus (WNV) has consistently been reported to be associated with human cases of illness in the region near Chicago, Illinois. However, the number of reported cases of human illness varies across years, with intermittent outbreaks. Several dynamic factors, including temperature, rainfall, and infection status of vector mosquito populations, are responsible for much of these observed variations. However, local landscape structure and human demographic characteristics also play a key role. The geographic and temporal scales used to analyze such complex data affect the observed associations. Here, we used spatial and statistical modeling approaches to investigate the factors that drive the outcome of WNV human illness on fine temporal and spatial scales. Our approach included multi-level modeling of long-term weekly data from 2005 to 2016, with weekly measures of mosquito infection, human illness and weather combined with more stable landscape and demographic factors on the geographical scale of 1000m hexagons. We found that hot weather conditions, warm winters, and higher MIR in earlier weeks increased the probability of an area of having a WNV human case. Higher population and the proportion of urban light intensity in an area also increased the probability of observing a WNV human case. A higher proportion of open water sources, percentage of grass land, deciduous forests, and housing built post 1990 decreased the probability of having a WNV case. Additionally, we found that cumulative positive mosquito pools up to 31 weeks can strongly predict the total annual human WNV cases in the Chicago region. This study helped us to improve our understanding of the fine-scale drivers of spatiotemporal variability of human WNV cases.

## Introduction

West Nile virus (WNV), a mosquito-borne zoonotic disease, was first identified in the United States in the summer of 1999 in New York City [1]. The mosquitoes of several *Culex* species

**Data Availability Statement:** Data cannot be shared publicly because of privacy concerns. Data are available from the Illinois Department of Public

Health Institutional Data Access / Ethics Committee, which has imposed these restrictions for researchers who meet the criteria for access to confidential data. Contact Megan Patel, Megan.Patel@illinois.gov (Surveillance and Informatics Epidemiologist in the Office of Health Protection, Division of Infectious Disease), for information about data access.

**Funding:** This publication was supported by Cooperative Agreement #U01 CK000505, funded by the Centers for Disease Control and Prevention (RLS). The funders had no role in study design, data collection and analysis, decision to publish, or preparation of the manuscript.

**Competing interests:** The authors have declared that no competing interests exist.

are the primary enzootic and bridge vectors for the transmission of WNV, and several bird species are known to contribute in the amplification of the virus [2–4]. Since its first successful invasion in New York, WNV quickly adapted to the local populations of *Culex* vector mosquitoes and avian populations and rapidly spread throughout the conterminous United States [5,6]. The first major WNV outbreak in the United States was observed in 2002, when more than 4,150 human cases and 284 deaths attributable to WNV infection were reported to the CDC from 40 states compared to only 149 cases and 19 deaths from 10 states cumulatively during the three years from 1999 to 2001 [7]. This stirred a prompt public health response from federal, state, and local public health agencies and led to the establishment of a more robust surveillance of mosquitoes and birds to monitor and control the spread of WNV [8].

Public health surveillance for West Nile virus (WNV) involves collection and testing of *Culex* vector mosquitoes, collection and testing of dead birds suspected to have died of WNV, testing of sentinel chickens or of wild birds captured for this purpose, and reporting of cases of human and equine illness [9]. The ultimate goal of these surveillance data is to target mosquito control, and thereby reduce illness through the reduction of the number of infected vector mosquitoes, and to target educational messages to warn citizens to reduce individual exposure. One additional advantage of having a strong surveillance system in place is that the long-term data generated can be integrated with publicly available weather, landscape, and socioeconomic data and can be used effectively to identify the important drivers of WNV transmission and to develop predictive models [10,11].

Several earlier studies have identified some of the important drivers of WNV transmission in humans. These factors include prior weather conditions and landscape structure that affect the mosquito's biological responses, the abundance and infection status of the vector mosquitoes, demographic and social characteristic of population, individual human behavior, and the level of public awareness [10–17]. For example, an analysis of 12 years of mosquito testing and human illness data in Ontario, Canada showed that, while the mosquito infection rate of one week earlier was the strongest temporal predictor of human risk of WNV, an epidemic threshold based on the cumulative positive *Culex* pools up to mid-August (week 34) can be successfully used to predict human WNV epidemics [16]. In Long Island, New York, more than 65% of forecast models based on past mosquito infection and human illness correctly predicted seasonal total human WNV cases up to 9 weeks before the first reported cases [18]. Similarly, the vector index, based on a combination of vector infection and abundance was found to be highly correlated with human WNV cases in studies conducted in Larimer County, Colorado (Fauver et al., 2015), and Dallas, Texas [19].

Weather factors are important drivers of WNV transmission due to their direct effect in mosquito biology. When compared with human WNV cases, higher than normal average annual temperatures are associated with an increased likelihood of higher WNV disease incidence, nationally and in most regions in the United States [17]. This relationship was true in Europe, too, where abnormally high July temperature was associated with higher incidence of human WNV cases [20]. The role of precipitation is often controversial and varies by study regions. For example, higher than normal precipitation was positively associated with higher human WNV cases in the eastern region of the United States, but this relationship was reversed for the western region [21]. Another study identified drought as an important driver of WNV epidemics in the United States [22]. Local landscape structures have also been associated with human WNV incidence. The important land cover variables associated with increased risks of human WNV include proximity to wetlands [23,24], higher tree density [24], irrigated and agricultural rural areas [25], urban areas characterized by higher impervious surfaces and storm sewer systems [26], and inner suburbs characterized by older houses, moderate vegetation and moderate population [27].

Apart from extrinsic factors, population structure, demographic characteristics, and individual variation also play roles in WNV epidemics [28]. As people age, especially when they have a history of hypertension and immunosuppression, their risk of WNV disease increases [29,30]. Community characteristics such as income level, the age of housing, management of sewer and drainage system, mosquito abatement practices, and public health infrastructure also determines the risk of WNV human infections [12,26].

Different spatial scales have been used in geographical analyses to identify the drivers of human risk from WNV infections. The most commonly used spatial scale in the United States is counties [17,22,31], census tracts or Zip Code Tabulation Areas (ZCTA) [12,32], census block groups [33], and buffers of varying sizes around trap locations or human cases [24]. Each of these spatial scales has its own inherent biases, as these political boundaries do not necessarily correspond to the ecological processes of the disease in question [34]. Alternatively, dividing the area into equal spaces, such as rectangular bins or hexagons, has been used to reduce some of these biases (e.g. [35]). Hexagonal grids have an additional advantage in that they reduce the edge effects, better fit curved surfaces, and have identical neighbors [36,37].

In Illinois, WNV human infections have been endemic since 2002, with annual variability in the number of cases [38]. The majority of the human WNV cases have been reported from the northeastern region, where the largest number of people in the state is congregated. A census tract level analysis in this region using human WNV occurrence data from the 2002 outbreak year identified that census tracts with lower population density, relatively close WNV positive dead bird specimens, a higher percentage of older white residents, and housing built between 1950 and 1959 were more likely to be associated with spatial clusters of WNV cases [12]. A follow up expanded this study to look at annual incidence of WNV human illness in northeastern Illinois from 2002 to 2006, with additional variables to assess the effects of rainfall, temperature and the WNV mosquito infection rate [39]. This analysis determined that white populations and housing from the 1950s were associated with increased illness in some years, but this was not consistent. Interestingly, census tracts with lower rainfall had higher rates of WNV illness, but the mosquito infection rate was not an important variable in any of the models [39].

Despite the identification of some of these potential risk factors, accurate prediction of human illness cases from WNV remains elusive at the local scale, especially as it is related to dynamic weather and mosquito infection status. Using long-term data on human WNV illness and intensive mosquito surveillance for the Chicago region, we can identify the fine scale drivers of spatiotemporal variability of human WNV epidemic in an urban environment. The overall goal of this study is to determine factors affecting the spatiotemporal variability of clinical WNV incidence in people through identification of the fine scale drivers of WNV transmission in an urban area with a repeated history of WNV outbreaks. These potential drivers include dynamic mosquito infection and weather. Our specific objectives in this study are to (i) describe the fine-scale temporal and spatial patterns of human WNV illness in the Chicago region, (ii) evaluate the temporal relationships between mosquito infection and human WNV illness, and (iii) determine the fine-scale dynamic effects of weather, land cover, mosquito infection, and demographic factors on the presence of human West Nile virus illness across time and space.

## Materials and methods

This project was approved by the Institutional Review Boards of the University of Illinois Urbana-Champaign and the Illinois Department of Public Health.

The two Illinois counties of Cook and DuPage, comprising Chicago and its suburbs, were included in this study. The total area covered by these two counties is nearly 5,100 square kilometers, and the total population in 2010 was 6.1 million. These areas were selected because of the relatively high incidence of human West Nile virus illness reported from these two counties and the long-term intensive mosquito surveillance data available for this region. The temporal window included in this study was the 24-week time period from late May to late October (weeks 22 to 45), which corresponds to the timing of mosquito activity and human WNV illness, with data for the years from 2005 to 2016. The years from 2002 to 2004, during which Illinois had its first invasion from WNV, were excluded in this analysis because of the absence of mosquito testing data. Data on avian and equid surveillance were not included as these programs were not consistently applied across the time period.

We chose to summarize all variables into hexagons to provide a neutral spatial unit of consistent size and shape, which is not possible with political boundaries. For this, we overlaid hexagons measuring 1000 m in diameter on the outlines of Cook and DuPage counties to create a grid of 5,345 hexagons for the study area. Out of these, 328 were excluded after a comparison with fine scale population data from the 2010 U.S. Census indicated that there were no households on record within those hexagons. Thus, 5,017 hexagons were included in the analysis. All independent variables related to weather, land cover, mosquito infection and demography were calculated for each hexagon, as described below.

## Mosquito data

Mosquito testing data from 2005 to 2016 were obtained from the Illinois Department of Public Health (IDPH) through a user agreement. The IDPH collates the data from local public health agencies and mosquito abatement districts across Illinois and maintains a statewide database for the results from WNV mosquito testing. The IDPH developed a mosquito surveillance protocol that local health and mosquito abatement districts are expected to follow in order to standardize the mosquito collection and testing across the state. In general, the local agencies collect vector mosquitoes with gravid traps, identify the sex and species of the mosquitoes, and make pools of up to 50 mosquitoes of a single species from those captured in each trap to test for the presence of WNV infection. When fewer then 50 mosquitoes are captured, a pool will consist of fewer than 50 mosquitoes. During the study period, the common tests used to identify WNV in mosquitoes included antigen assays, VecTest or the Rapid Analyte Measurement Platform (RAMP) test. Some pools were also tested by Real Time reverse transcriptase polymerase chain reaction (RT-PCR). In instances when a pool was tested using more than one type of test, only the RT-PCR results were used in the analysis. Our analysis used only the test results from pools of female *Culex* mosquitoes. Not all mosquitoes were identified to species prior to testing; however, the majority of *Culex* collected in this region belong to the species *Cx. pipiens* or *Cx. restuans* [3].

To determine the location of the mosquito traps, we used the existing latitude and longitude recorded in the IDPH database. In cases where the spatial data were missing, we geocoded the trap locations based on the address provided. Our analysis used all the trap locations recorded from 2005 to 2016 from Cook and DuPage counties in addition to any traps located within a 10 km radius from their boundaries (located within Lake, McHenry, Kane, Kendall, and Will counties). For each trap, the mosquito infection rate (MIR) was calculated by week and by year using the formula $1000 * \frac{\text{number of positive pools}}{\text{total number of mosquitos in pools tested}}$ [40].

Using MIR calculations from all traps, we developed continuous surface maps for MIR for each week and year using the inverse distance weighting (IDW) interpolation technique in ArcGIS 10.1. From this interpolated surface map for each year and week, the average,

minimum, and maximum MIR for each hexagon was calculated using the zonal statistics as table function in ArcGIS 10.1. A model builder platform using iteration features in ArcGIS 10.1 was used to run these processes.

## Human illness data

Records of human WNV cases in Illinois were obtained from the IDPH through a user agreement. All confirmed and probable cases of WNV reported to the IDPH by medical and public health personnel for the study area were included in this study; the state of Illinois mandates reporting of WNV to local public health departments, which then report all cases to IDPH. Probable cases are those that meet clinical criteria during the season when transmission is likely to occur and meet laboratory criteria for West Nile virus by serology (IgM capture ELISA) or polymerase chain reaction, while confirmed cases are those with confirmatory test results from the IDPH or the Centers for Disease Control and Prevention. All the human WNV cases in Cook and DuPage counties reported from 2005 to 2016 were geocoded and aggregated by hexagons for each week and year. The data were converted into the binary form of presence or absence of a WNV case in a given hexagon and week.

## Demographic data

The demographic variables included were total population, racial composition, housing age, and income level. The total population and racial composition included the number of White, African American, Asian, and Hispanic people at the census block level, as reported in the 2010 U.S. Census. The racial population data was converted to the percentage of White, African American, Asian, and Hispanic people in each hexagon. The income data for the block group level were obtained from the 2015 American Community Survey. Housing age was included as the proportions of housing built in different time periods, which was obtained at the block group level from the 2015 American Community Survey. We divided housing age into four different time-periods: pre-World War II houses (built before 1939), post-World War II houses (built between 1940 and 1969), houses built between 1970 and 1989, and houses built after 1990. These demographic data were processed in ArcGIS using the intersection tool to calculate a parameter for each hexagon.

## Landcover data

Landcover data for the entire United States was obtained from the national landcover database (NLCD) for the years 2006 and 2011. The NLCD database is a Landsat based landcover data available at a 30 m resolution (www.mrlc.gov). The landcover raster was clipped for Cook and DuPage counties, including a surrounding 1 km buffer. From this clipped raster, the total number of pixels for each land category within each hexagon was calculated using the tabulate as area tool in ArcGIS 10.1. The proportion of each land cover category for each hexagon was then calculated by dividing the number of pixels for that category by the total number of pixels for all categories. In Cook and DuPage counties, 15 different types of landcover were available: urban areas (developed open space, developed low intensity, developed medium intensity, developed high intensity), forests (deciduous, evergreen and mixed), barren land, shrubs, grassland, pasture, cultivated crops, woody wetlands, herbaceous wetlands, and open water. The land cover data from 2006 was used to analyze the WNV cases for the years from 2006 to 2010, while the land cover data from 2011 was used for 2011 to 2016.

## Weather data

Spatial weather data on daily mean temperature and precipitation from 2005 to 2016 were obtained from the PRISM Climate Group (PRISM Climate Group, Oregon State University, http://prism.oregonstate.edu). The PRISM daily data are available as spatial grids of 4 km resolution, which are calculated through interpolation and statistical techniques using point data from weather monitoring networks across the country combined with topographic data. These daily data were used to calculate the weekly temperature and precipitation. For our analysis, the weekly mean temperature was calculated by taking the average of the seven daily averages for that week, and the weekly precipitation was calculated as a sum of the daily precipitation for that week. Finally, the weekly temperature and precipitation for each year and week for each hexagon was calculated by using the zonal statistics as table function in ArcGIS 10.1. We also calculated average January temperature for each hexagon for each year from the daily data as a proxy for the winter temperature.

## Statistical methods

To assess the temporal relationship between human illness and MIR, we calculated the Spearman rank correlation between the weekly MIR of 1–6 weeks lag and human cases. We repeated this analysis on the subsets of years with high numbers of WNV cases (more than 100 human cases; 2005, 2006, 2012 and 2016) and those with low numbers of WNV cases (less than 100 cases; 2007–2009, 2010, 2011, 2013–2015) to examine if the relationship between MIR and human cases varies in high and low years. We further examined the ability of the early summer (weeks 22–27) and mid-summer (weeks 28–33) average MIR to explain and predict the seasonal annual total WNV cases by using linear regression analysis. In addition, we assessed the ability of the cumulative positive mosquito pools up to week 28 and thereafter, added to each week's data, to find a threshold that could best explain the annual total human WNV cases. In both of these calculations, data from 2005 to 2014 were used to create a regression equation, and data from 2015 and 2016 was used to test the model.

To visualize the spatial patterns of human illness over time, we first developed choropleth maps of WNV cases. Then, we used local Moran's I method using an inclusive second order queen contiguity weight matrix in the spatial analysis software GeoDa to further identify the spatial clusters of cumulative human WNV cases from 2005 to 2016. We also examined differences in results using neighboring cells and rook contiguity weight matrix, but the results did not vary.

For the spatiotemporal statistical model, the outcome variable was the presence/ absence of a human WNV case in each hexagon for each year and week. The predictors included 32 variables related to weather, land cover, mosquito infection and demography (Table 1). The weather variables consisted of mean weekly temperature and precipitation with lags of one to four weeks. The land cover variables include 15 categories, the proportion for each hexagon of: developed open space; developed low, medium, and high intensity urban areas; deciduous, evergreen, and mixed forests; barren land; shrubs; grassland; pasture; cultivated crops; woody wetlands; herbaceous wetlands; and open water. The mosquito infection data included the average MIR with lags of one to four weeks for each hexagon for each year and week. Demographic variables for each hexagon included the proportion of White, African American, Asian, and Hispanic population and the average median household income. In total, there were 1.44 million rows of data (5017 hexagons * 12 years * 24 weeks). A correlation matrix among all variables was created to evaluate multicollinearity before running the model. As our response variable was binary (presence or absence of WNV human cases), we used mixed effects multiple logistic regression with stepwise selection for the statistical analysis, with

**Table 1. List of explanatory variables.**

| Variables | Notation |
|---|---|
| *Land cover* | |
| Proportion of developed open space | dospct |
| Proportion of developed low intensity | dlipct |
| Proportion of developed medium intensity | dmipct |
| Proportion of developed high intensity | dhipct |
| Proportion of deciduous forests | dfpct |
| Proportion of evergreen forests | efpct |
| Proportion of mixed forests | mfpct |
| Proportion of barren land | blpct |
| Proportion of shrubs | shrubspct |
| Proportion of grassland | glandpct |
| Proportion of pasture | pasturepct |
| Proportion of cultivated land | clpct |
| Proportion of woody wetlands | wwpct |
| Proportion of herbaceous wetlands | hwpct |
| Proportion of open water | owpct |
| *Mosquito infection rate* | |
| Mosquito infection of one week before | mirlag1 |
| Mosquito infection of two weeks before | mirlag2 |
| Mosquito infection of three weeks before | mirlag3 |
| Mosquito infection of four weeks before | mirlag4 |
| *Weather* | |
| *Temperature* | |
| Average temperature of one week before | templag1 |
| Average temperature of two weeks before | templag2 |
| Average temperature of three weeks before | templag3 |
| Average temperature of four weeks before | templag4 |
| *Precipitation* | |
| Average precipitation of one week before | precilag1 |
| Average precipitation of two weeks before | precilag2 |
| Average precipitation of three weeks before | precilag3 |
| Average precipitation of four weeks before | precilag4 |
| *Demographic factors* | |
| Percentage of White population | whitepct |
| Percentage of African American | blackpct |
| Percentage of Asian population | asianpct |
| Percentage of Hispanic | hispanicpct |
| Median household income | Income |

hexagons as a random variable. We used the PROC GLIMMIX procedure in the SAS statistical software. An Akaike information criterion (AIC) was used to choose the best model [41]. A receiver operating characteristics (ROC) curve was calculated using model predictions for 2015 and 2016 to evaluate model performance. All the statistical analyses were conducted in SAS 9.4 (SAS Institute Inc., Cary).

**Table 2. Annual human WNV cases, average seasonal mosquito infection rate (MIR), and mosquito testing from 2005 to 2016 in Cook and DuPage counties.**

| Year | Number of human cases | Average MIR | Number of pools tested | Number of positive pools | Total number of mosquitoes tested |
|------|----------------------|-------------|------------------------|--------------------------|-----------------------------------|
| 2005 | 181 | 5.33 | 7,165 | 1,939 | 271,235 |
| 2006 | 129 | 5.35 | 9,428 | 1,984 | 318,386 |
| 2007 | 43 | 2.65 | 12,131 | 1,259 | 375,520 |
| 2008 | 10 | 1.91 | 9,024 | 587 | 298,995 |
| 2009 | 1 | 1.14 | 9,450 | 298 | 311,220 |
| 2010 | 47 | 5.19 | 11,491 | 2,086 | 393,279 |
| 2011 | 24 | 3.10 | 8,911 | 939 | 287,774 |
| 2012 | 229 | 7.35 | 10,162 | 3,182 | 323,497 |
| 2013 | 66 | 4.26 | 11,078 | 1,967 | 407,326 |
| 2014 | 31 | 2.97 | 9,273 | 990 | 333,489 |
| 2015 | 36 | 3.57 | 7,725 | 1,046 | 314,363 |
| 2016 | 108 | 6.34 | 6,144 | 1,687 | 219,909 |

MIR = Mosquito infection rate; WNV = West Nile virus

## Results

There were 1,371 total human WNV cases reported in Illinois from 2005 to 2016. Out of these total reported cases, 906 cases (66%) were from the Chicago region (Cook and DuPage Counties). The number of human WNV cases in the study region varied annually, with the year 2012 reporting the highest number of cases (229) and the lowest number of cases (1) reported in 2009 (Table 2). The average annual MIR during the mosquito season was also highest in 2012 (7.34), with 31.3% of tested pools positive for WNV (Table 2). The number of mosquito pools tested annually ranged from about 6,000 pools in 2016 to over 12,000 pools in 2007.

We found a strong temporal relationship between the MIR of previous weeks and human WNV cases in the study region (Table 3, Fig 1). The strongest correlation (r = 0.837) was with MIR at a one-week lag (Table 3). The strength of the correlation was stronger (r = 0.884) in the subset of high infection years (2005, 2006, 2012, and 2016) and relatively lower (r = 0.737) in low years (Table 3). When evaluated for only 2012, when case counts were highest, the correlation between MIR and human WNV cases was also the highest (r = 0.899). In both high and low years, the strength of the correlation gradually declined with the number of weeks lagged and there was almost no correlation with MIR after lags of four weeks.

**Table 3. Spearman correlation of weekly cumulative human WNV cases and lagged MIR for all years and selected subsets of years from 2005–2016 in Cook and DuPage counties.**

| | | Years with Human WNV cases | | |
|---|---|---|---|---|
| MIR | All years | >100 | <100 | 229 (Year 2012) |
| Same week | 0.776 | 0.775 | 0.671 | 0.818 |
| One week before | 0.837 | 0.884 | 0.737 | 0.899 |
| Two weeks before | 0.765 | 0.766 | 0.698 | 0.875 |
| Three weeks before | 0.601 | 0.574 | 0.556 | 0.727 |
| Four weeks before | 0.429 | 0.354 | 0.394 | 0.501 |
| Five weeks before | 0.289 | 0.147 | 0.286 | 0.283 |
| Six weeks before | 0.142 | 0.001 | 0.120 | 0.038 |

MIR = Mosquito infection rate; WNV = West Nile virus

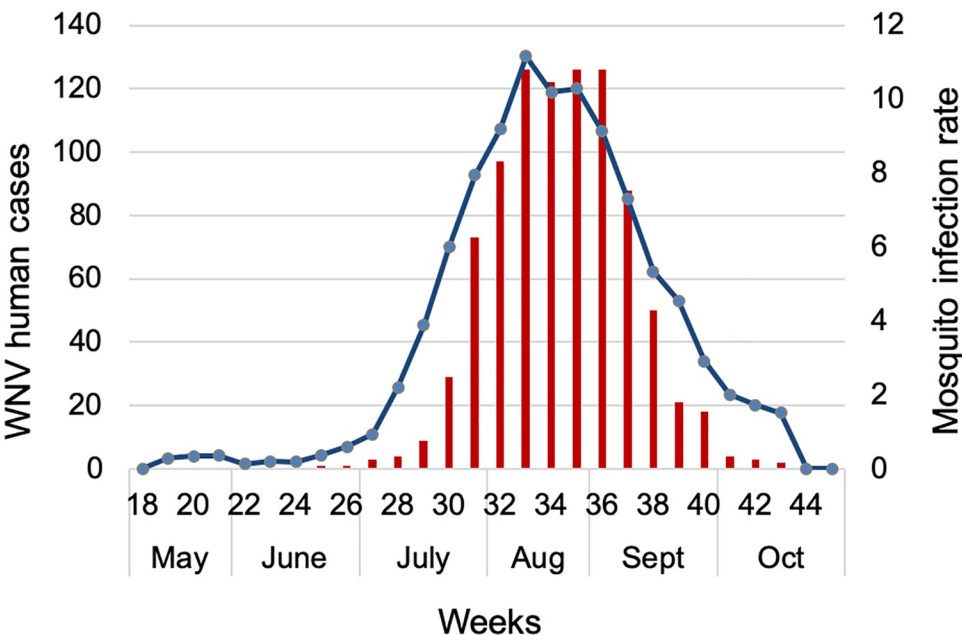

**Fig 1. Cumulative weekly human WNV cases (red bars) and mosquito infection rate (blue line) from 2005–2016 in Cook and DuPage counties, Illinois.**

We found that the MIR of mid-summer (weeks 28–33) was able to explain 93% of the variability in total annual human cases (Table 4, Fig 2). The model predicted 44.8 human cases for 2015, compared to 35 actual cases, and 142.7 human cases for 2016 compared to 108 actual cases. Likewise, the cumulative number of positive pools also strongly explained and predicted the total annual human cases (Table 4, Fig 3); the cumulative number of positive mosquito pools by week 31 explained 93% of the variability in total annual human cases, similar to that explained by mid-summer MIR (Table 4). The model using cumulative positive pools by week 31 predicted 35.1 human cases (vs. 35 actual cases) for 2015 and 102.8 human cases (vs. 108

**Table 4. The regression equations of the relationship between a cumulative number of WNV positive pools, mosquito infection rate in a six-week period early and mid-summer and human West Nile virus illnesses for the year for Cook and DuPage counties from 2004–2014.**

| Week | Regression equation | R-square |
|---|---|---|
| 28 | 30.1 + 0.445 * Number of positive pools | 0.721 |
| 29 | 21.2 + 0.278 * Number of positive pools | 0.825 |
| 30 | 11.8 + 0.194 * Number of positive pools | 0.895 |
| 31 | 2.33 + 0.144 * Number of positive pools | 0.931 |
| 32 | - 6.0 + 0.118 * Number of positive pools | 0.917 |
| 33 | - 16.5 + 0.103 * Number of positive pools | 0.901 |
| 34 | - 23.7 + 0.0938 * Number of positive pools | 0.863 |
| 35 | - 29.5 + 0.0861 * Number of positive pools | 0.813 |
| Early summer (22–27) | 13.7 + 162 * average MIR of week 22–27 | 0.833 |
| Mid-summer (28–33) | - 16.7 + 14.7 * average MIR of week 28–33 | 0.936 |

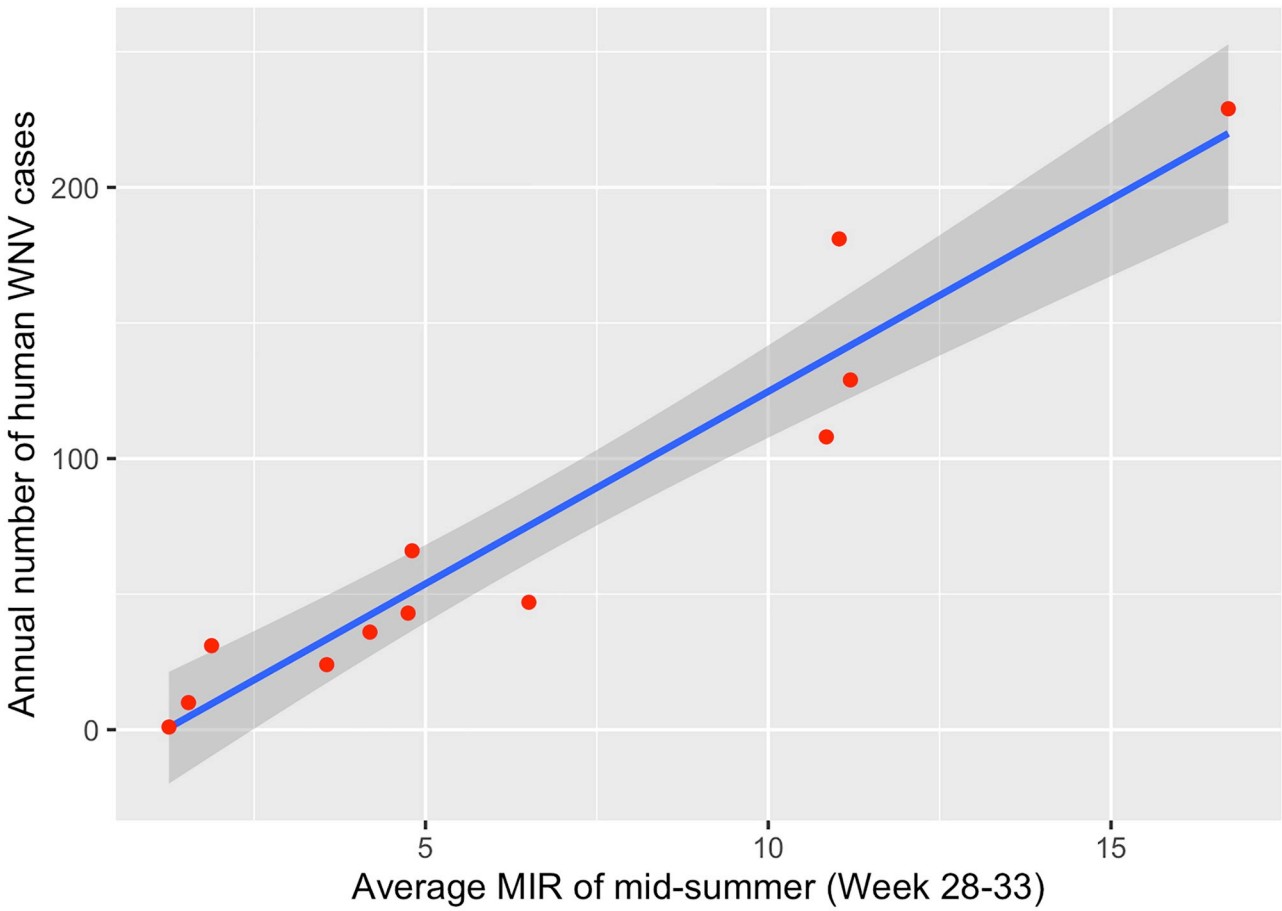

**Fig 2. The relationship between annual human WNV infections and mid-summer mosquito infection rate from 2005–2014 in Cook and DuPage counties, Illinois.** Blue line indicates the best-fit linear regression line, with gray shading indicating 95% confidence intervals. Red points are observations (one per year).

actual cases) for 2016. The cumulative mosquito positive pools by week 31 thus better predicted the annual human cases than the mid-summer MIR.

The spatial pattern of human WNV cases in Cook and DuPage counties showed that cases were distributed throughout most areas of the study region at some point during the study period, with some pockets of higher numbers of cases (Fig 4). Out of the total 5,345 hexagons in the study area, 750 hexagons had experienced at least one case of human WNV case during the years 2005 to 2016. Cumulatively, 123 hexagons had more than one human WNV case, with the maximum number of cases in a hexagon being five (Fig 4). The local Moran's I identified some spatial clusters of human WNV cases in Cook and DuPage counties (Fig 5): 92 hexagons with higher numbers of cases were also near to others with higher numbers of cases.

The results of the mixed-effects regression analysis showed that temperature, precipitation, land cover, mosquito infection, and demographic characteristics are all associated with the probability of an area having a case of WNV human illness. The AIC criteria used to compare the 10 best competing models showed that a model consisting of 15 variables that included temperature, MIR, land cover, and demographic characteristics was the best model (S1 Table). The final multivariable model indicated that higher temperatures two, three, and four weeks

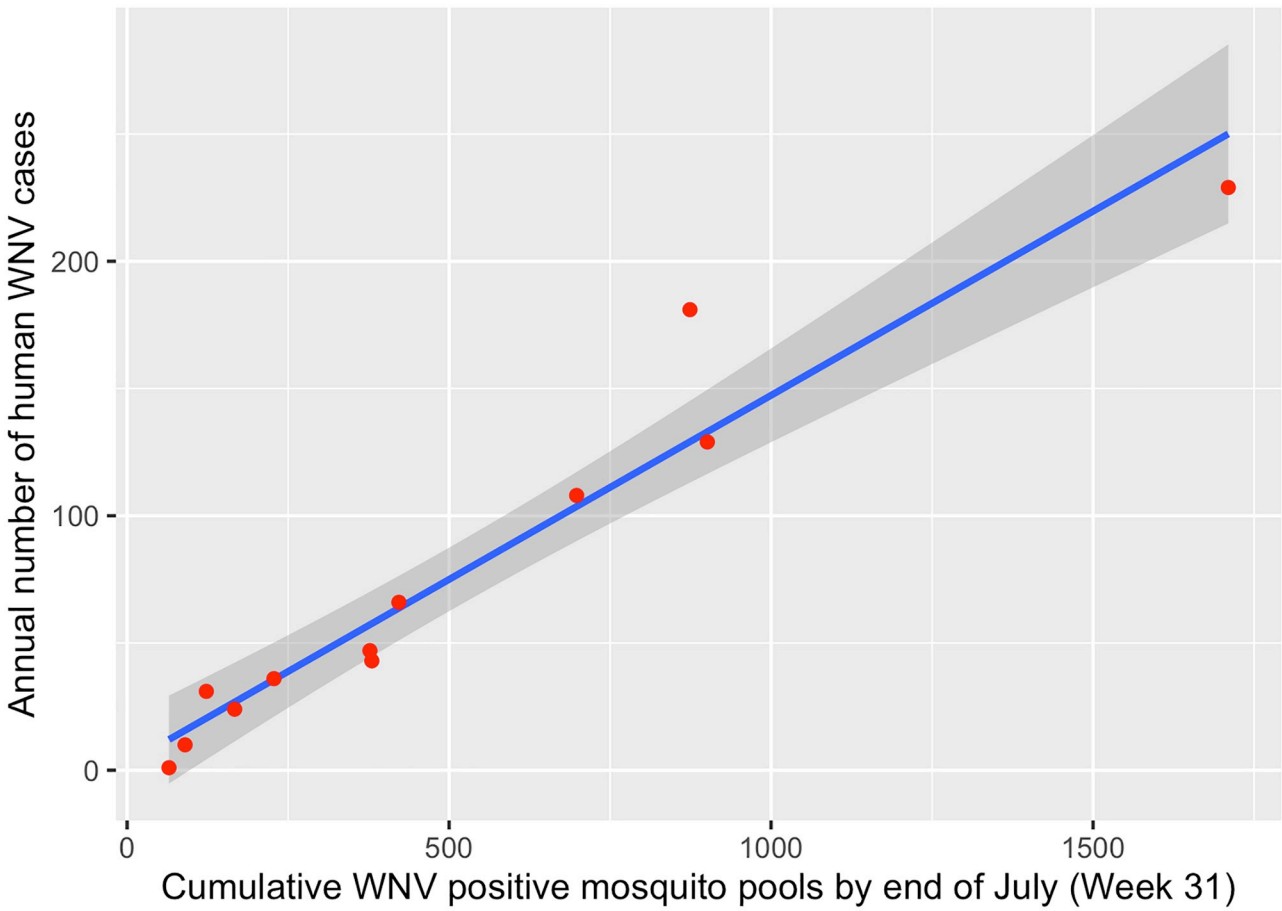

**Fig 3. The relationship between annual human WNV infections and a cumulative number of WNV positive mosquito pools from 2005–2014 in Cook and DuPage counties, Illinois.** Blue line indicates the best-fit linear regression line, with gray shading indicating 95% confidence intervals. Red points are observations (one per year).

earlier and warmer average January temperature were associated with a higher probability of a hexagon being positive for human WNV case (Table 5). The lagged mosquito infection rates of one to four weeks earlier were also positively associated with the outcome variable (Table 5). Among the land cover variables, the proportion of open water, grassland, and deciduous forests were negatively associated with the probability of a WNV case while the proportion of low intensity developed areas was positively associated (Table 5). Among the demographic variable, total population was found to be positively associated with the probability of a WNV case, while the proportion of housing built after 1990 was negatively associated (Table 5). The area under the ROC curve was 0.948, which indicates that model performance was excellent (Fig 6).

## Discussion

We identified important fine-scale drivers of spatiotemporal variability in the human WNV cases in Chicago region, Illinois, an area of ongoing WNV transmission. Our analysis used long-term data on human illness, mosquito surveillance, weather, landscape, and demographic data. We found significant spatial clusters of human WNV cases within this urban

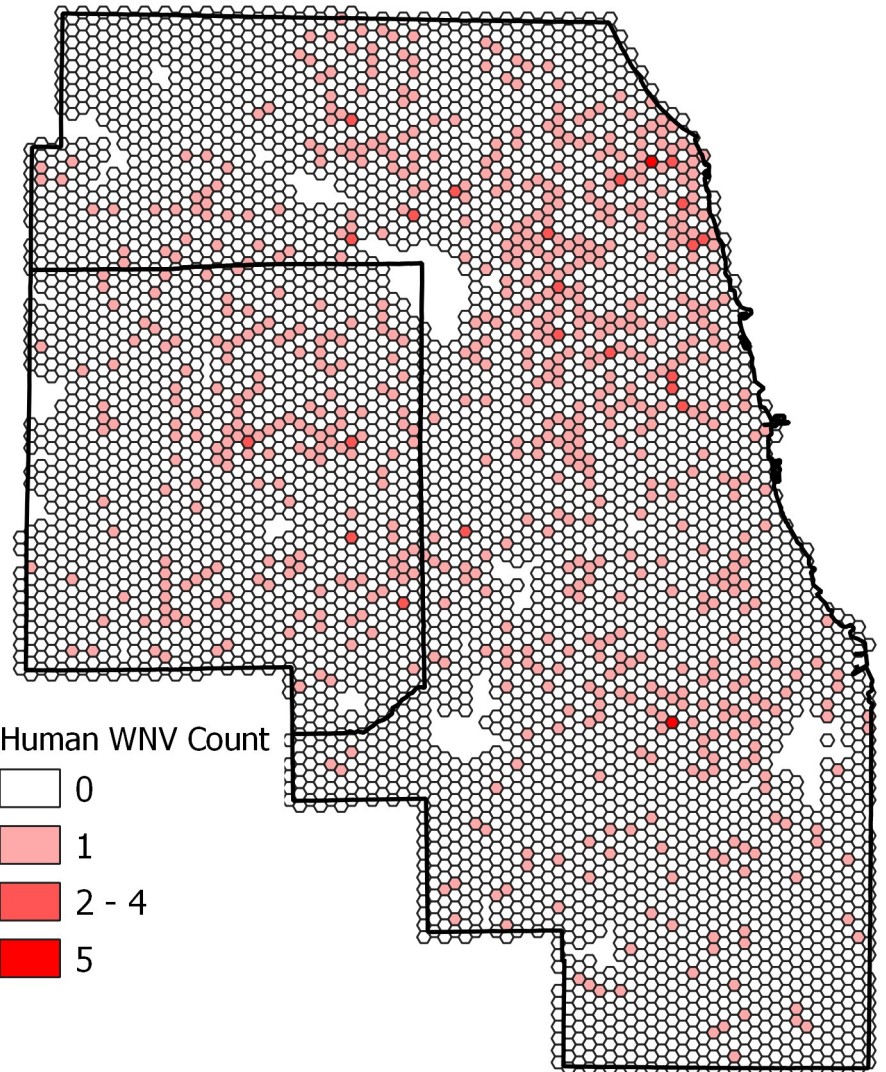

**Fig 4. The spatial distribution of the cumulative number of human WNV infections from 2005–2016 in Cook and DuPage counties, Illinois.** Colors indicate the number of human cases.

environment. We also found a strong correlation between the weekly MIR of earlier weeks and weekly human WNV cases, and further developed predictive temporal models using mid-summer average MIR and cumulative positive mosquito pools which can be used to estimate the total annual human WNV cases.

The temporal variation in the weekly human WNV cases was strongly correlated with MIR of one to four weeks earlier, with a correlation of one week earlier being the strongest. This finding was similar to our earlier model based on Illinois climate divisions, in which Division 2 includes our current study area [42]. The similarity in the correlation may be due to the fact that the data for Climate Division 2 were dominated by the data from Cook and DuPage, as these counties have more intensive surveillance compared to other Illinois counties. However, similar observations were also found in Ontario, Canada, where MIR of one week earlier was most strongly correlated with the weekly variation in human WNV cases [16]. In our study,

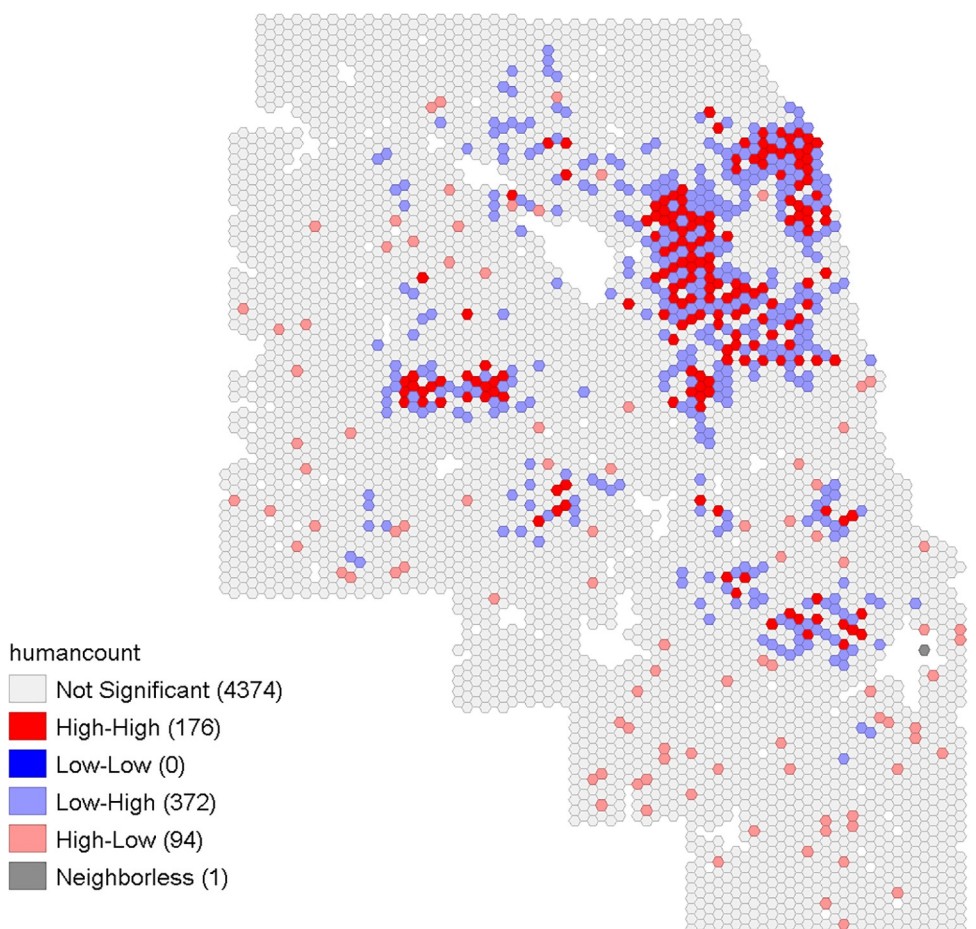

**Fig 5. The local Moran's I result showing the spatial clustering of cumulative human WNV infections from 2005–2016 in Cook and DuPage counties, Illinois.** Value ranges for not significant are -1.209 to 3.278, high-high are 0.86 to 13.96, low-high are -0.854 to -0.175, and high-low are -2.252 to -0.654.

we also found that the correlations between weekly MIR and human cases increased in high WNV years, which was also observed in a study conducted in Long Island, New York [18]. This is understandable, as stochastic variability decreases with increased numbers of cases, allowing for more precise estimation.

The temporal models we developed using mid-summer average MIR and cumulative mosquito positive pools were both able to explain more than 90% of the variability in the annual number of human cases. This similarity of the results was not surprising, as positive mosquito pools are used to calculate the MIR. However, the cumulative positive pools up to week 31 better predicted the annual human cases compared to mid-summer average MIR for 2015 and 2016. The difference observed between the two approaches may reflect the variability of the MIR calculation depending on the mosquito pool size [43,44]. Taking the most extreme possibility, when there was only one mosquito in a pool and it tested positive, this would yield a MIR of 1000 in contrast to MIR of 20 when a pool with 50 mosquitoes was tested positive. In Ontario, Canada, the cumulative number of positive mosquito pools up to week 34 was suggested as an action threshold potential to estimate the total annual human cases [16]. In

**Table 5. Model parameters for the best model using weather, land cover, mosquito infection, and demographic factors to predict the occurrence of WNV human cases in Chicago region.**

| Variable | Parameter estimate | F-value | P-Value | Odds ratio (95% CI) |
|---|---|---|---|---|
| Fixed effects | | | | |
| Year | - | 17.33 | <0.001 | - |
| Temperature of two weeks before | 0.06963 | 22.36 | <0.001 | 1.08 (1.049–1.112) |
| Temperature of three weeks before | 0.1085 | 42.99 | <0.001 | 1.128 (1.092–1.165) |
| Temperature of four weeks before | 0.1628 | 116.47 | <0.001 | 1.197 (1.162–1.234) |
| Average January temperature | 0.3613 | 16.65 | <0.001 | |
| Mosquito infection rate of one week before | 0.003199 | 21.53 | <0.001 | 1.003 (1.002–1.004) |
| Mosquito infection rate of two weeks before | 0.003938 | 38.79 | <0.001 | 1.004 (1.002–1.005) |
| Mosquito infection rate of three weeks before | 0.004003 | 37.83 | <0.001 | 1.004 (1.002–1.005) |
| Mosquito infection rate of four weeks before | 0.003958 | 34.63 | <0.001 | 1.004 (1.002–1.005) |
| Total population | 0.000225 | Infinity | <0.001 | 1.009 (1.006–1.012) |
| Open water percentage | -0.05527 | 9.58 | 0.002 | 0.954 (0.921–0.988) |
| Developed light intensity percentage | 0.01848 | 80.65 | <0.001 | 0.990 (0.985–0.994) |
| Deciduous forest percentage | -0.02401 | 4.66 | 0.0309 | 0.985 (0.980–0.991) |
| Grassland percentage | -0.04603 | 3.14 | 0.0763 | |
| Post 1990 built housing percentage | -0.00546 | 4.28 | 0.0386 | |
| Random effect | | | | |
| Subject | Estimate | Standard error | Z-value | P-value |
| Hexagon ID | 1.1769 | 0.1636 | 7.19 | <0.0001 |

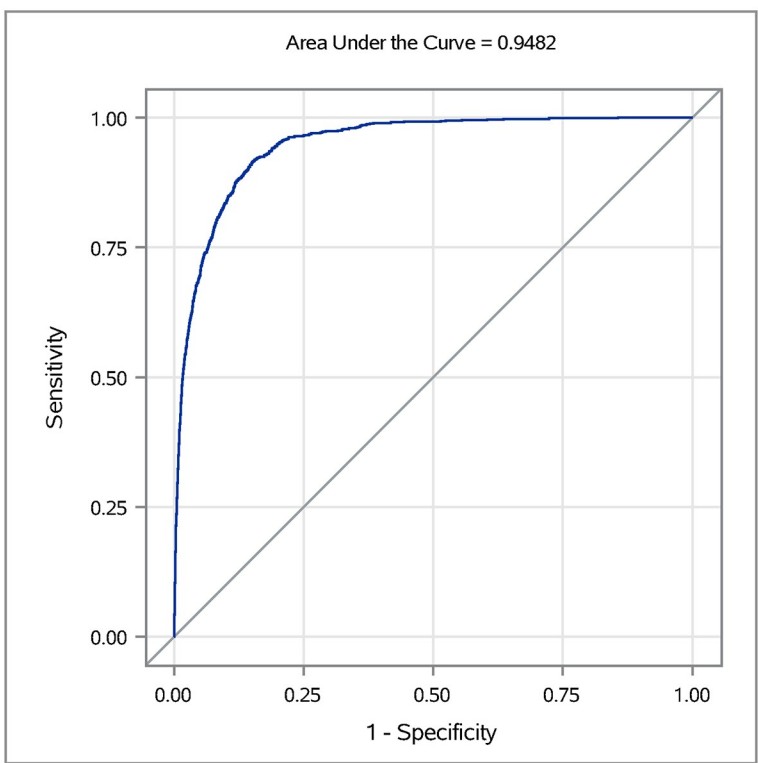

**Fig 6. The receiver operating characteristics (ROC) curve for the final best-fit model for human West Nile virus cases in the Chicago area, based on weather, land cover, mosquito infection, and demographic factors.**

Chicago, we obtained this signal three weeks earlier, which can be crucial to the ability to intervene in the upcoming potential human WNV outbreak.

We found spatial clustering of human WNV cases within the study area, indicating that some areas were more likely than others to have a WNV human case. A spatial clustering pattern of human WNV cases in Chicago area was also observed in the 2002 WNV outbreak year [12]. Several factors might play a role in the observed spatial clustering pattern, including differences in the fine-scale variation in the local landscape structure that affects mosquito population, fine-scale weather variation, demographic characteristics, access of people to health care system, and spatially variable mosquito abatement practices [12,39,45,46].

In this study, through multilevel modeling, we identified several dynamic factors that are possibly driving the fine scale spatiotemporal variation in the human WNV cases occurrence in the Chicago region. We found that the higher temperature in the previous weeks increases the probability of an area being positive for a WNV case. The association between higher temperature and WNV human illness has also been observed in other studies conducted at different spatial scales [15,17,20]. This is possibly due to the dynamic effect of higher temperature on mosquito breeding and virus replication [35,47–49]. The unique feature of our study is that by considering the dynamic nature of weather, we allowed the temperature and precipitation to vary both temporally and spatially to capture the better role of weather in the spatiotemporal variability of human WNV cases. The precipitation of earlier weeks was not as important as the temperature of the preceding weeks but still was moderately important. The negative association of precipitation observed indicated that dry and hot weather conditions would increase the probability of an area being positive for a WNV case. Some other studies have also indicated that hot dry weather conditions are conducive for WNV transmission [50,51]. While it may seem counter-intuitive that the proportion of open water was negatively correlated with WNV cases, given *Culex* populations would increase with an increase in breeding sites, the definition of open water (areas in which any aquatic vegetation is submerged) is such that it is unlikely to provide good breeding habitat for *Culex*.

We also found increased MIR up to four weeks earlier will increase the probability of an area being positive for a WNV human case. The temporal association between lagged MIR and human WNV cases is relatively well established [10,16,52]. However, it was interesting to find the positive association of MIR when spatiotemporal variabilities of human cases were considered. In our current analysis, we found that areas with a higher percentage of white population had a higher probability of being positive for WNV, which has also been observed in a previous study of this [12]. This may be a function of access to the health care system and likelihood of seeking medical treatment and testing [12,27], or may simply be due to high proportions of white population in areas of the study region where environmental conditions are also conducive to increased mosquito activity.

This study also found that the probability of a hexagon being a positive for WNV case decreased in developed medium and high-intensity urban areas and increased in developed low-intensity urban areas, indicating that the suburban areas of Chicago are more at risk than the highly developed urban centers. The lack of mosquito breeding grounds and bird activity in the high-intensity urban areas might be responsible for this. Previous studies conducted in the same area have also indicated that sub-urban region in Chicago is at more risk from the WNV [12,27]. This is probably due to the poor sanitation system in the older houses compared to new houses.

In this study, we did not consider prior seasonal differences in the weather conditions, which we recommend be incorporated in future studies. In addition, the calculation of MIR for hexagons may be biased as the IDW interpolation technique used to develop continuous surface maps is affected by the uneven distribution of mosquito traps across the study area.

Alternatively, other interpolation methods such as kriging might be used to develop continuous surface maps for MIR, as this method takes into account spatial autocorrelation and also creates an error map. In this study, we did not distinguish between neuroinvasive and non-neuroinvasive WNV cases. Separate analysis for only neuroinvasive cases might help us to identify what conditions drive the occurrence of the severe form of WNV infection and should also help to reduce diagnostic bias. Also, in future studies, we might consider using different spatial scales to identify if the geographic scale has affected the results. We were also unable to use data from avian or equid surveillance in this study, despite its usefulness in other modeling approaches [53–55], due to the lack of consistent data across the time period. Bird surveillance in Illinois is limited to passive surveillance of a small number of dead birds tested in each county per year, and is generally suspended after WNV is known to be circulating in the area, while equid surveillance is based entirely on passive self-reporting [56]. This lack of consistent data on avian mortality has been noticed by others [10], and remains an issue for the use of data on the primary host in WNV forecasting.

In conclusion, our analysis helped to better understand the fine-scale dynamic drivers of WNV transmission in an urban environment. The dynamic interplay between temperature and precipitation, mosquito infection, land cover, and demographic characteristics determine the probability of an area having a WNV case or not. Additionally, we established an important temporal relationship between cumulative mosquito positive pools and mid-summer average MIR with the total annual human WNV cases. This information can be used as a guideline to develop a threshold for public health intervention.

## Supporting information

**S1 Table. Candidate models for predicting the probability of human WNV occurrence using weather, land cover, mosquito infection, and demographic factors in Chicago region.** (DOCX)

## Acknowledgments

The contents of this publication are solely the responsibility of the authors and do not necessarily represent the official views of the Centers of Disease Control and Prevention or the Department of Health and Human Services.

## Author Contributions

**Conceptualization:** Surendra Karki, Marilyn O'Hara Ruiz.

**Data curation:** Surendra Karki, William M. Brown, Marilyn O'Hara Ruiz.

**Formal analysis:** Surendra Karki, John Uelmen, Marilyn O'Hara Ruiz.

**Funding acquisition:** Marilyn O'Hara Ruiz.

**Methodology:** Surendra Karki, John Uelmen.

**Project administration:** Rebecca Lee Smith.

**Resources:** William M. Brown.

**Supervision:** Marilyn O'Hara Ruiz, Rebecca Lee Smith.

**Visualization:** Surendra Karki, William M. Brown, John Uelmen, Marilyn O'Hara Ruiz, Rebecca Lee Smith.

**Writing – original draft:** Surendra Karki, Marilyn O'Hara Ruiz, Rebecca Lee Smith.

**Writing – review & editing:** Surendra Karki, William M. Brown, John Uelmen, Rebecca Lee Smith.

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
