## [Decision Letter · Decision Letter 0]

17 Feb 2020

PONE-D-19-34216

The drivers of West Nile virus human illness: fine scale dynamic effects of weather, mosquito infection, social, and biological conditions

PLOS ONE

Dear Dr. Smith,

Thank you for submitting your manuscript to PLOS ONE. After careful consideration, we feel that it has merit but does not fully meet PLOS ONE’s publication criteria as it currently stands. Therefore, we invite you to submit a revised version of the manuscript that addresses the points raised during the review process.

Comments of both reviewers should be addressed. In addition, please elaborate more to render Figure captions self explained by including all respective information. Also, the references list should be checked carefully and editing mistakes should be corrected. For example all scientific names at species and genus level should be given in italics. Last table 5 could be included as supplementary materials.

We would appreciate receiving your revised manuscript by Apr 02 2020 11:59PM. To enhance the reproducibility of your results, we recommend that if applicable you deposit your laboratory protocols in protocols.io, where a protocol can be assigned its own identifier (DOI) such that it can be cited independently in the future. For instructions see: http://journals.plos.org/plosone/s/submission-guidelines#loc-laboratory-protocols

We look forward to receiving your revised manuscript.

Kind regards,

Nikos T Papadopoulos

Academic Editor

PLOS ONE

Journal Requirements:

3. Your ethics statement must appear in the Methods section of your manuscript. If your ethics statement is written in any section besides the Methods, please move it to the Methods section and delete it from any other section. Please also ensure that your ethics statement is included in your manuscript, as the ethics section of your online submission will not be published alongside your manuscript.

Reviewers' comments:

Reviewer's Responses to Questions

**Comments to the Author**

1. Is the manuscript technically sound, and do the data support the conclusions?

Reviewer #1: Yes

Reviewer #2: Yes

2. Has the statistical analysis been performed appropriately and rigorously? 

Reviewer #1: Yes

Reviewer #2: I Don't Know

3. Have the authors made all data underlying the findings in their manuscript fully available?

Reviewer #1: Yes

Reviewer #2: Yes

4. Is the manuscript presented in an intelligible fashion and written in standard English?

Reviewer #1: Yes

Reviewer #2: Yes

5. Review Comments to the Author

Reviewer #1: Peer review report on PLOS ONE manuscript " The drivers of West Nile virus human illness: fine scale dynamic effects of weather, mosquito infection, social, and biological conditions", (Manuscript number PONE-D-19-34216).

Recommendation: Minor Revision

Comments to Authors:

This manuscript analyzes the available long-term data of mosquito infection rates, West Nile virus human cases and weather variables from 2005 to 2016 combined with landscape and demographic characteristics of two Illinois counties of the Chicago region in order to evaluate relationships between the factors on fine temporal and spatial scale and identify the drivers that potentially affect the presence of human WNV illness and may act as early warning predictors.

The paper is well written with a well-organized text, the data were analyzed using multi-level statistical modeling approaches and the findings are sufficiently documented and the results are valuable for a better understanding of the fine-scale drivers of spatiotemporal variability of WNV human case prevalence in an urban environment such as in the study area.

Although numerous published studies that have shed light on factors that affect WNV transmission in an area, the knowledge regarding the influence of climatic variables in correlation with the data from the entomological surveillance and the number of WNV human cases, is still limited.

For that reason, the paper makes a substantial contribution to the literature and is therefore recommended for publication in PLOS-ONE after minor revision taking into account the following general or specific comments.

General comments

The study uses and analyzes the 10-year data (2005 to 2014) from Cook and DuPage counties in the Chicago, Illinois region and the accuracy of the predictions of the developed model tested with the data of the same specific area.

However, according to the literature, it is well known that models predicting the WNV transmission and human WNV infections do not always have the same accuracy when applied to other areas with different mosquito fauna, weather conditions and/or geomorphological and demographic characteristics. Therefore, we consider that the study area should also be mentioned in the title.

Please comment and, if necessary, provide an adequate justification in the manuscript, for the reason that in this work were note included data from passive or active monitoring of WNV presence in birds and equids, which are considered by several authors as important prediction factors of the presence and spread of WNV virus in an area.

Specific comments

Line 170 of the manuscript: If available, please provide information on the species of Culex mosquitoes that have been tested for WNV presence as the vectorial competence of different species may vary significantly for WNV transmission to humans.

Line 179 of the manuscript: Please add a bibliographical reference in the reference section for the MIR estimation tool by Biggerstaff, 2006.

Line 188 of the manuscript: Please provide a definition and some additional information about the category of "probable cases of WNV" that were also included in the study along with the "confirmed cases" because the symptoms of infection by the West Nile vary in severity, with the mild forms can be easily confused with flu symptoms and usually go unreported.

Lines 578-580 of the manuscript: Please, correct Reference no 39 by adding the name of the journal, volume number and pages numbers.

Messina JP, Brown W, Amore G, Kitron UD, Ruiz MO. West Nile Virus in the Greater Chicago Area : A Geographic Examination of Human Illness and Risk from 2002 to 2006. URISA Journal 2011;23: 5-18.

Reviewer #2: Dear authors,

This is a well written paper that deals with the determination of factors affecting the spatiotemporal variability of WNV cases in humans through identification of the fine scale drivers of WNV transmission in an urban area with a repeated history of WNV outbreaks. The findings are very interesting since they include multi-level modeling of weekly data from over a decade and they extend our knowledge in the correlation of variables related to temperature, precipitation, mosquito infection, land cover, and demographic characteristics with the probability of an area having a WNV case or not.

Further down please consider some comments of minor importance that may benefit the manuscript.

It seems that the infection status of avian population, as primary reservoirs of WNV, and equids, as dead-end hosts, were not included among the tested variables for modeling structure. Please note that these are critical factors implicated in the WNV transmission in order to develop predictive models.

As mentioned in the introduction, public health surveillance for WNV involves collection and testing of dead birds suspected to have died of WNV, testing of sentinel chickens or of wild birds captured for this purpose and reporting of cases of equine illness.

Could you please justify this data gap in the model structuring? Is there any surveillance system for infected avian and equids population in the study area?

In the “Introduction” you may add any relevant literature data where bird and/or equine infection rate were used for development of models predicting WNV transmission in humans. Also, in lines 440-459 of the manuscript, you could mention the fact that avian and equids infection status was not considered as a factor for prediction of WNV cases in humans in the study area.

According to the best multivariable model that was used, the proportion of open water was negatively associated with the probability of WNV cases. Also, as mentioned in the discussion, a negative association of precipitation and WNV cases was observed and this indicates that dry and hot weather conditions would increase the probability of an area being positive for a WNV case.

Instead, it is supposed that high rainfall and high percentage of water bodies in an area may favor mosquito population by increasing their breeding sites, and therefore may lead to increased WNV cases in humans. Hence, a positive correlation between precipitation and water bodies with WNV cases in humans is anticipated. Please comment.

6. PLOS authors have the option to publish the peer review history of their article (what does this mean?). If published, this will include your full peer review and any attached files.

Reviewer #1: No

Reviewer #2: No

---

## [Author Response · Author response to Decision Letter 0]

27 Feb 2020

Reviewer #1: Peer review report on PLOS ONE manuscript " The drivers of West Nile virus human illness: fine scale dynamic effects of weather, mosquito infection, social, and biological conditions", (Manuscript number PONE-D-19-34216).

Recommendation: Minor Revision

Comments to Authors:

This manuscript analyzes the available long-term data of mosquito infection rates, West Nile virus human cases and weather variables from 2005 to 2016 combined with landscape and demographic characteristics of two Illinois counties of the Chicago region in order to evaluate relationships between the factors on fine temporal and spatial scale and identify the drivers that potentially affect the presence of human WNV illness and may act as early warning predictors.

The paper is well written with a well-organized text, the data were analyzed using multi-level statistical modeling approaches and the findings are sufficiently documented and the results are valuable for a better understanding of the fine-scale drivers of spatiotemporal variability of WNV human case prevalence in an urban environment such as in the study area.

Although numerous published studies that have shed light on factors that affect WNV transmission in an area, the knowledge regarding the influence of climatic variables in correlation with the data from the entomological surveillance and the number of WNV human cases, is still limited.

For that reason, the paper makes a substantial contribution to the literature and is therefore recommended for publication in PLOS-ONE after minor revision taking into account the following general or specific comments.

• Thank you for your comments and your feedback!

General comments

The study uses and analyzes the 10-year data (2005 to 2014) from Cook and DuPage counties in the Chicago, Illinois region and the accuracy of the predictions of the developed model tested with the data of the same specific area.

However, according to the literature, it is well known that models predicting the WNV transmission and human WNV infections do not always have the same accuracy when applied to other areas with different mosquito fauna, weather conditions and/or geomorphological and demographic characteristics. Therefore, we consider that the study area should also be mentioned in the title.

• Thank you for the suggestion, we have made that change

Please comment and, if necessary, provide an adequate justification in the manuscript, for the reason that in this work were note included data from passive or active monitoring of WNV presence in birds and equids, which are considered by several authors as important prediction factors of the presence and spread of WNV virus in an area.

• We have added a statement (145-147) that the avian and equid surveillance programs were not consistent across the time period, and added a discussion section (467-474) about the point.

Specific comments

Line 170 of the manuscript: If available, please provide information on the species of Culex mosquitoes that have been tested for WNV presence as the vectorial competence of different species may vary significantly for WNV transmission to humans.

• We agree that is an important point; we have added some information as to common species in the region.

Line 179 of the manuscript: Please add a bibliographical reference in the reference section for the MIR estimation tool by Biggerstaff, 2006.

• Thank you, corrected

Line 188 of the manuscript: Please provide a definition and some additional information about the category of "probable cases of WNV" that were also included in the study along with the "confirmed cases" because the symptoms of infection by the West Nile vary in severity, with the mild forms can be easily confused with flu symptoms and usually go unreported.

• We have added the information. The difference between probable and confirmed cases is confirmatory testing by either IDPH or CDC; all cases had positive diagnostic results and clinical signs during the likely transmission season.

Lines 578-580 of the manuscript: Please, correct Reference no 39 by adding the name of the journal, volume number and pages numbers.

Messina JP, Brown W, Amore G, Kitron UD, Ruiz MO. West Nile Virus in the Greater Chicago Area : A Geographic Examination of Human Illness and Risk from 2002 to 2006. URISA Journal 2011;23: 5-18.

• Corrected

Reviewer #2: Dear authors,

This is a well written paper that deals with the determination of factors affecting the spatiotemporal variability of WNV cases in humans through identification of the fine scale drivers of WNV transmission in an urban area with a repeated history of WNV outbreaks. The findings are very interesting since they include multi-level modeling of weekly data from over a decade and they extend our knowledge in the correlation of variables related to temperature, precipitation, mosquito infection, land cover, and demographic characteristics with the probability of an area having a WNV case or not.

• Thank you

Further down please consider some comments of minor importance that may benefit the manuscript.

It seems that the infection status of avian population, as primary reservoirs of WNV, and equids, as dead-end hosts, were not included among the tested variables for modeling structure. Please note that these are critical factors implicated in the WNV transmission in order to develop predictive models.

As mentioned in the introduction, public health surveillance for WNV involves collection and testing of dead birds suspected to have died of WNV, testing of sentinel chickens or of wild birds captured for this purpose and reporting of cases of equine illness.

Could you please justify this data gap in the model structuring? Is there any surveillance system for infected avian and equids population in the study area?

In the “Introduction” you may add any relevant literature data where bird and/or equine infection rate were used for development of models predicting WNV transmission in humans. Also, in lines 440-459 of the manuscript, you could mention the fact that avian and equids infection status was not considered as a factor for prediction of WNV cases in humans in the study area.

• We have added a statement as to the inconsistent application of avian and equid surveillance in this region (145-147), and given more information about that surveillance in the discussion (467-474), including references to models using these data types.

According to the best multivariable model that was used, the proportion of open water was negatively associated with the probability of WNV cases. Also, as mentioned in the discussion, a negative association of precipitation and WNV cases was observed and this indicates that dry and hot weather conditions would increase the probability of an area being positive for a WNV case.

Instead, it is supposed that high rainfall and high percentage of water bodies in an area may favor mosquito population by increasing their breeding sites, and therefore may lead to increased WNV cases in humans. Hence, a positive correlation between precipitation and water bodies with WNV cases in humans is anticipated. Please comment.

• Open water is classified as areas in which any aquatic vegetation is submerged, as opposed to woody or herbaceous wetlands. This is not likely to be stagnant water of the type used by Culex mosquitoes for breeding. Therefore, the negative association between proportion of open water and WNV cases is most likely due to the fact that open water, as defined, does not favor the mosquito population. We have noted this in the discussion (434-438).

---

## [Editor Report · Decision Letter 1]

27 Mar 2020

PONE-D-19-34216R1

The drivers of West Nile virus human illness  in the Chicago, Illinois, USA area : fine scale dynamic effects of weather, mosquito infection, social, and biological conditions

PLOS ONE

Dear Dr. Smith,

Thank you for submitting your manuscript to PLOS ONE. After careful consideration, we feel that it has merit but does not fully meet PLOS ONE’s publication criteria as it currently stands. Therefore, we invite you to submit a revised version of the manuscript that addresses the points raised during the review process.

The paper is much improved and there no need for additional review besides some important editorial elements that need to be addressed. 

**Please my previous comments as far as improvements in figure legends and in references list are regarded.**

**" **.... In addition, please elaborate more to render Figure captions self explained by including all respective information. Also, the references list should be checked carefully and editing mistakes should be corrected. For example all scientific names at species and genus level should be given in italics. Last table 5 could be included as supplementary materials."

**Please consider the attached annotated pdf and also make some additional proofreading especially in reference. **

We would appreciate receiving your revised manuscript by May 11 2020 11:59PM. To enhance the reproducibility of your results, we recommend that if applicable you deposit your laboratory protocols in protocols.io, where a protocol can be assigned its own identifier (DOI) such that it can be cited independently in the future. For instructions see: http://journals.plos.org/plosone/s/submission-guidelines#loc-laboratory-protocols

We look forward to receiving your revised manuscript.

Kind regards,

Nikos T Papadopoulos

Academic Editor

PLOS ONE

---

## [Author Response · Author response to Decision Letter 1]

3 Apr 2020

In addition, please elaborate more to render Figure captions self explained by including all respective information. 

 Completed

Also, the references list should be checked carefully and editing mistakes should be corrected. For example all scientific names at species and genus level should be given in italics. 

 Corrected

Last table 5 could be included as supplementary materials.

 Done

Please consider the attached annotated pdf and also make some additional proofreading especially in reference.

 All corrections have been made

---

## [Editor Report · Decision Letter 2]

10 Apr 2020

The drivers of West Nile virus human illness  in the Chicago, Illinois, USA area : fine scale dynamic effects of weather, mosquito infection, social, and biological conditions

PONE-D-19-34216R2

Dear Dr. Smith,

We are pleased to inform you that your manuscript has been judged scientifically suitable for publication and will be formally accepted for publication once it complies with all outstanding technical requirements.

With kind regards,

Nikos T Papadopoulos

Academic Editor

PLOS ONE
---

## [Editor Report · Acceptance letter]

15 Apr 2020

PONE-D-19-34216R2 

The drivers of West Nile virus human illness  in the Chicago, Illinois, USA area : fine scale dynamic effects of weather, mosquito infection, social, and biological conditions 

Dear Dr. Smith:

I am pleased to inform you that your manuscript has been deemed suitable for publication in PLOS ONE. Congratulations! Your manuscript is now with our production department. 

With kind regards,

on behalf of

Dr. Nikos T Papadopoulos 

Academic Editor

PLOS ONE